# Focused Strategies for Defining the Genetic Architecture of Congenital Heart Defects

**DOI:** 10.3390/genes12060827

**Published:** 2021-05-28

**Authors:** Lisa J. Martin, D. Woodrow Benson

**Affiliations:** 1Division of Human Genetics, Cincinnati Children’s Hospital Medical Center, Cincinnati, OH 45229, USA; 2Department of Pediatrics, University of Cincinnati School of Medicine, Cincinnati, OH 45229, USA; 3Department of Pediatrics, Medical College of Wisconsin, Wauwatosa, WI 53226, USA; dbenson@mcw.edu

**Keywords:** cardiovascular malformations, genetic variation, association, etiology, phenotyping

## Abstract

Congenital heart defects (CHD) are malformations present at birth that occur during heart development. Increasing evidence supports a genetic origin of CHD, but in the process important challenges have been identified. This review begins with information about CHD and the importance of detailed phenotyping of study subjects. To facilitate appropriate genetic study design, we review DNA structure, genetic variation in the human genome and tools to identify the genetic variation of interest. Analytic approaches powered for both common and rare variants are assessed. While the ideal outcome of genetic studies is to identify variants that have a causal role, a more realistic goal for genetic analytics is to identify variants in specific genes that influence the occurrence of a phenotype and which provide keys to open biologic doors that inform how the genetic variants modulate heart development. It has never been truer that good genetic studies start with good planning. Continued progress in unraveling the genetic underpinnings of CHD will require multidisciplinary collaboration between geneticists, quantitative scientists, clinicians, and developmental biologists.

## 1. Introduction

The objective of this review is to provide an overview of strategies utilized in pursuit of defining the genetic origins of rare birth defects characterized by malformation of the heart, so-called congenital heart defects (CHD). CHD occurs during heart development, also known as cardiogenesis, which begins very early in gestation. The initial beating embryonic heart is tubular-shaped, and all the while functioning to support somatic growth of the fetus, the embryonic heart loops and is morphed to a complex four-chambered organ [1]. Identification of the regulatory networks controlling all stages of cardiogenesis has led to improved understanding of genes involved in heart development [2]. Conversely, complimentary, genetic studies of CHD patients have identified variants in genes essential to heart development [3]. We hope that review of focused strategies will facilitate studies that improve our understanding of the genetic architecture of human CHD.

## 2. Phenotypic Considerations

### 2.1. What Is CHD?

CHD refer to malformations of the heart present at birth. We will not consider malformations incompatible with fetal life but will limit our focus on CHD compatible with fetal development resulting in live birth. The anatomic details and clinical significance of the malformations are quite varied. Some may have a profound impact on postnatal clinical well-being and require intervention, usually surgical and often in the neonatal period. On the other hand, some CHD may have no clinical impact and go undetected until discovered incidentally later in life. While such defects may be of little clinical significance, they are highly relevant to the design of genetic studies. This is because statistical evaluation of the co-occurrence between genotype and phenotype sought in a genetic study requires determination of the presence or absence of CHD.

### 2.2. What Is the Incidence of CHD?

CHD constitute a major portion of clinically significant birth defects. While an incidence of 10 per 1000 (~1%) is often cited [4], depending on the definition of what constitutes CHD, the incidence may be much higher. A history of clinically significant CHD, such as those requiring surgery, may be readily apparent during review of past medical history. However, some CHD may have been overlooked because they were not clinically significant; they cause no symptoms and may only be detected by a cardiac imaging study. Examples of such CHD include isolated aneurysm of the atrial septum, persistent left superior vena cava (LSVC), right aortic arch and bicuspid aortic valve (BAV). BAV, the most common congenital cardiac malformation, has an incidence of 10 to 20 per 1000 of the population [5]; BAV and other malformations of little apparent clinical significance are often excluded from estimates of CHD incidence. Taken together, CHD incidence may be as high as 50 per 1000 (~5%) [6]. This is an important consideration in genetic study design because even though CHD in a research participant lacks clinical significance it may be an indication of genetic abnormality and thereby be of profound genetic significance.

### 2.3. Tools to Determine CHD Phenotype

Classification of CHD phenotype is based on careful consideration of images of the heart position in the thorax, heart chambers, septa and valves as well as location, anatomy and relationships of vena cava, pulmonary veins and great arteries. A chest X-ray may be useful for determining the position of the heart in the thorax. Angiography, an invasive procedure, may provide useful images of abnormal cardiac anatomy. However, an echocardiogram which uses ultrasound technology has become the gold standard technique for clinical cardiac imaging as it is well adapted for cardiac imaging for patients of all ages including the fetus.

Details of pre- and postnatal medical history, family history and clinical exam may inform about the presence or absence of CHD; in genetic discovery studies, identification of the absence of CHD is as important as discovering the presence of CHD. Individuals with CHD may have a history of cardiac surgery, previous visits to a cardiologist, and/or records of past cardiac imaging, e.g., echocardiography, that reveal the CHD phenotype. However, the absence of such records does not equate to a phenotype of normal. Extracardiac features relevant to a CHD diagnosis include facial features, skeletal abnormalities including malformed vertebrae and limb anomalies, and abdominal viscera arrangement. For example, from the perspective of an echocardiographer, tetralogy of Fallot due to del22q11, also known as DiGeorge syndrome, may be indistinguishable from that CHD due to the NKX2.5 mutation, whereas in the same scenario a history of cleft palate would drastically alter the genetic focus [3]. A developmental assessment including gross and fine motor skills as well as cognitive development may lead to recognition of developmental delay which is more likely to be associated with certain CHD as part of a syndrome.

Family history can distinguish genetic conditions that are not usually inherited, e.g., Down syndrome (trisomy 21), from genetic conditions that exhibit familial clustering, e.g., BAV. The recognition of familial heart disease has been complicated by several genetic phenomena (Table 1) that obscure the familial nature [7]. Further, while most individuals believe family history is important, many are unfamiliar with important, relevant clinical details of familial CHD. Too often, in the hustle and bustle of a busy clinic, family history is asked on the initial visit, recorded and never revisited. This leads to a situation whereby family history is an under-utilized tool in the recognition of genetic etiology. Family history is dynamic, and a current account may require revisiting the questions on more than one occasion and obtaining information from more than one family member. A pedigree is a shorthand way to document and record family history and may give some indication as to the mode of inheritance. However, with electronic medical records, pedigrees may be attached as images that can be reviewed manually and updated as necessary.

A genetic condition may be identified by recognizing signature cardiac and/or noncardiac findings during evaluation. For example, tetralogy of Fallot is a signature cardiac malformation for 22q11 deletion syndrome (del22q11), but a physician evaluating a patient with right ventricular outflow tract malformation may overlook dysmorphic facial features characteristic of del22q11. The presence of syndromic features is strongly supportive of a genetic condition and may be an indication for genetic testing. Even with what appears to be isolated CHD, typical features of the cardiac phenotype may suggest a genetic etiology with known inheritance.

## 3. Genetic Considerations

### 3.1. What Is DNA?

Deoxyribonucleic acid (DNA) carries the information necessary to create and organize all cells and organs in the human body (Figure 1). DNA is a two-stranded molecule with a backbone made of sugar and phosphate. Attached to each sugar is one of four bases known as nucleotides (adenine, cytosine, guanine, and thymine). The strands are held together by bonds between the bases. The base sequences carry the information not only on how to make proteins but also when to make proteins. An individual’s DNA carries complete information from each parent, and the paired set of information is known as the genome. 

In humans, DNA is organized into 23 chromosomes which harbor three billion base pairs (bp). Genes are the DNA units that make proteins; the human genome has ~30,000 genes. Only ~1% of the genome codes for protein. The DNA sequences encoding these genes are known as exons (Figure 1). Most genes have multiple exons, with the interspersing region known as introns. Following transcription and RNA splicing, introns are removed, and exons join to form a contiguous coding sequence.

The sequences near an intron-exon boundary are critical for appropriate splicing to occur. While early work on the genome focused on the exons or protein coding regions, it is well recognized that DNA of the non-protein coding regions plays an important role in phenotypic variation among individuals [8,9]. For example, the regions upstream of the gene include the promotor sequence which contain the transcription initiation site which determines whether the DNA is copied to messenger RNA (the first step in creating a protein). Additionally, upstream and downstream elements may influence transcription [10]. Notably, except when copying itself, DNA is stored in a compact form wound around nucleosomes. As such, some regulatory elements may be long distances from the gene of interest [11]. While much of the focus on CHD genetics has been on variants which affect protein structure, work such as ENCODE [12] has demonstrated the importance of the non-protein coding sequences as well. Indeed, a recent paper by Gelb and colleagues demonstrated enrichment of de novo (not inherited from parents) variants in non-coding regions in CHD patients [13].

### 3.2. Types of Genetic Variation

Changes (variation) in DNA can be classified according to size and type of variation (Table 2). Small-scale variation (<1 kilobase, kb) includes single nucleotide variants (SNV), short insertions or deletions (indels) and repetitive elements (RE; e.g., tandem repeats and transposable elements). Large structural variants (SV) involve DNA segments >1kb and include deletions, duplications, insertions, inversions, translocations, and copy number variation (CNV). Genetic variation is the rule rather than the exception; it is responsible for the phenotypic differences between individuals and may be beneficial or harmful. Genetic variation alters the phenotype by modifying (i) protein coding sequences, (ii) promoters and other regulatory elements, (iii) splice sites and other regions affecting transcript structures, and (iv) other genomic regions with unknown direct connections to known protein function. As new genetic techniques have become available, numerous examples of genetic variation have been identified in CHD patients (reviewed in [3]). Taken together, the study of genetic variation has improved our understanding of the underpinnings of CHD.

### 3.3. Tools to Identify Genetic Variation

Generally, the tools can be divided into two types: those that seek variation within the human genome and those that are targeted to specific genomic locations. While the karyotype has been widely used in the past century to identify numerous types of SV [14], in this paper we will focus on more recently developed high-throughput technologies. In the late 1990s and early 2000s, microarray genotyping chips were developed to capture known SNVs [15]. Notably, these genotyping chips capture only a fraction of the SNV variation; however, the seminal International HapMap project demonstrated that only a fraction of the variants would be required to capture common variation due to linkage disequilibrium patterns [16]. Beyond SNV, these chips have been used to detect CNV [17], including in clinical laboratories [18]. 

**Table 2 genes-12-00827-t002:** Characteristics of types of genetic variation.

Type of Variant	Description	Consequence	Laboratory Methods	Examples in CHD
Single Nucleotide Variation (SNV)	Substitution of single bp for another bp	Individuals harbor ~3 million SNV [19,20]. Many have no known functional effect. Can alter protein structure or regulation	Array +++ NGS +++ LRS +++	Commonly identified in BAV and HLHS patients, e.g., NOTCH1 [21,22], GATA4 [23], GJA1 [24], and LRP2 [25]
Small Insertion/ Deletion (indel)	1–50 bp duplicated or deleted	Multiple mechanisms of mutagenesis possible	NGS ++ LRS +++	SMAD4 [26] ETS1 [27]
Tandem Repeats (TR)	Repeats (1–100 bp) occur at single locus	Present in at least 1/3 of human protein sequence [28]. Contribution to disease [29,30,31] related to alterations of gene expression [32].	LRS +++	Emerging area of focus. The number of repeats in Fragile X associated with cardiovascular outcomes [33].
Transposable Elements (TE)	Repeats (100 bp–20 kbp) occur at multiple loci	Common, accounting for more than 1/3 of the mammalian genome. ● Can impact protein structure [34] or gene regulation [35,36,37,38].	LRS +++	Emerging area of focus.
Copy Number Variation (CNV)	Duplication or deletion covering 1 Kb or greater.	Major source of human genome variation [39]. May alter gene expression [40,41]. Plant studies suggest that up to 50% overlap genes/ gene regulatory regions [42]. Account for up to 1 in 8 CHD cases [3,43].	Array ++ NGS ++ LRS +++	Aneuploidies, trisomies, and large SV are often associated with CHD [44,45]. Rare CNV are enriched in both BAV and HLHS [46,47,48,49,50].

+ = effectiveness of method to capture.

Another option to capture genetic variation is next generation sequencing (NGS). Briefly, NGS reads millions of small fragments of DNA in parallel [51]; these reads must be pieced back together using bioinformatics tools, which include quality control, alignment to a reference genome, and variant calling [52,53]. NGS can be used to capture SNV and some indels as well as CNV. NGS can be targeted to specific genomic regions, capture the exomes (WES), or capture the whole genome (WGS). The benefit of WES is a major reduction in costs (both from the laboratory perspective as well as data processing and storage). It has been shown that WES provides less even and more biased coverage than WGS (Figure 1) [54,55]. SV detection using WES is still a challenging area [55]. Finally, the NGS approaches are not assessable to all areas of the human genome, thus some variation will be missed [56].

Whereas NGS can produce reads up to 600 bp, long-read sequencing (LRS) routinely captures reads in excess of 10,000 bp (Figure 1), originating from single DNA molecules. Current technologies include Single Molecule Real Time (SMRT) by Pacific Biosciences of California (PacBio, Menlo Park, CA, USA) and Nanopore by Oxford Nanopore Technologies (ONT, Oxford, UK) [57]. As noted above, in NGS the small fragments have to be realigned using a reference genome, which is comprised of multiple individuals but regions which capture a single individual can create bias [58]. LRS is amenable to de novo assembly resulting in fewer missed regions [59] and identification of more structural variation [60,61,62]. Further, LRS is the only high throughput technology to effectively capture RE [32]. A major challenge for LRS is that it requires ultra-long, high molecular weight DNA; thus, specialized DNA isolation protocols applied to fresh sample or intact cells is necessary [57]. It is important to note that the bioinformatic processes for LRS differ markedly from NGS approaches. While both SMRT and ONT are excellent at capturing structural variation, only SMRT LRS capture SNV and indels with high accuracy [63,64].

## 4. Design and Analytic Considerations

### 4.1. Strategies to Identify Genetic Variants of Interest

The goal of genetic studies is to find variants that contribute to the etiology of the phenotype of interest, in our case CHD. The initial steps in this process are i) identify the study subjects, ii) specify the technique to capture genetic information, and iii) specify the analytic approach to establish genetic variants of interest. When designing a study, it is important to consider analytic approaches for both common and rare variants [65]. However, it is important to remember that analytic evidence, in and of its own, is not sufficient to demonstrate that a variant causes CHD. Further, demonstrating that using independent participants to identify the same variants, e.g., replication, supports the confidence that a variant may contribute to CHD. However, given the correlated structure of the genome, replication should not be considered sufficient to implicate a variant in CHD etiology.

Types of study design considerations are summarized in Table 3. At the crux of all genetic discovery studies is the collection of study subjects (Case Type) and specification of a method (Study Design) to distinguish affected from unaffected (Control Type) subjects. As described above, the decision related to phenotype definition is not trivial. Inclusion of too broad a phenotype may introduce heterogeneity in the underlying etiology, thereby reducing power. However, based on animal models and human studies, CHD exhibits phenotypic plasticity [66,67,68,69]; thus, too narrow a phenotype may miss individuals with shared etiology. Additionally, most CHD studies do not consider extra-cardiac features. While on one hand the extra-cardiac features may be part of the underlying etiology, e.g., pleiotropy, it may also be simply by chance that an individual has causal variants for CHD as well as causal variants for extra-cardiac features. When studying unrelated cases, disentangling phenotypic plasticity and pleiotropy can be challenging. However, studying multiple affected individuals from the same family may alleviate these challenges. For example, for phenotypic plasticity, one can study how a broader phenotype segregates with a variant [66,68]. Further, for pleiotropy, family studies can allow researchers to determine how often the two traits co-occur and the number of variants of interest shared in common [70].

The selection of controls is also critical as biases in controls may impact power to identify variants. Ideally, controls should be (i) phenotyped using similar protocols to the cases, (ii) genotyped with the same platform at the same time as the cases, and (iii) have a similar ancestral background (at minimum race and ethnicity) to the cases. Above, we emphasized the importance of appropriately phenotyping the cases. It is essential that controls are phenotyped equivalently to the cases. Failure to phenotype controls means that affected individuals may be included as a control; this will reduce power. While CHD is considered a rare condition, some phenotypes may go undetected in “normal, healthy” individuals. For example, BAV occurs in 1–2% of the population and may be unrecognized short of a cardiac imaging study. Thus, for optimal study power, it is essential that controls are as rigorously phenotyped as cases. Ideally, differences in genotype are due to phenotype differences between cases and controls and not, for example, because of variation in laboratory procedures, e.g., differences in genotyping chips, sequencing platforms, or genotyping chip version. For sequencing-based data, the version (build) of the human genome to which the data is aligned is also critical as not all regions are captured equally across all builds [71]. Matching of controls based on ancestry is essential as ancestral differences between cases and controls can cause spurious association [72]. Ancestral mismatching can be identified by evaluating the distribution of test statistics across the genome, the genomic inflation factor [73,74]. If mismatching is occurring, principal components analyses can be used to identify cases and controls with similar background genetic composition.

There are several sources of controls. Local controls recruited from the same geographic area as the cases are likely to be similar in ancestral distribution. Unfortunately, out of study controls who do not meet these criteria are commonly used. For example, genetic data from millions of individuals is available to researchers through portals such as dbGAP [75,76]. Although their use as out of study controls can markedly reduce the cost of genetic studies, dissimilarity of ancestral background may introduce noise that obscures key findings. Lastly, there are family-based controls. Unaffected parents are often included in studies of rare conditions to help exclude variants not contributing to disease. In extended family designs, multiple unaffected family members may be included. For rare conditions, the inclusion of extended family members may yield a large number of unaffected individuals. In these cases, sequential sampling where all first-degree relatives are phenotyped, and only when additional family members are identified, is the pedigree expanded. The combinations of types of cases and controls establishes the study design (Table 3).

### 4.2. Analytic Approaches

Analytic approaches can be separated into those which are powered for common variants (present in at least 1% of the population) or those optimal for rare variants (present in less than 1% of the population). We will first describe common variant approaches followed by a discussion of rare variants. For common variants, several assumptions should be evaluated prior to analysis. First, the variant should be evaluated for deviations from Hardy Weinberg Equilibrium (HWE). For variants which are not exhibiting selective pressure, deviations from HWE often are the result of erroneous genotypes [77]. Second, the genetic composition of the cases and controls should be evaluated by using principal component analyses [78] and the genomic inflation factor [73,74]. Differences in the genetic composition can be accounted for by selection of comparable groups (genetic matching) or by inclusion of principal components as covariates in the analyses [79]. To minimize risk for population stratification, most genetic analyses are stratified by continental ancestry, e.g., European, Asian, or African. For studies which evaluate variants across the genome, the risk of false positive association is high unless appropriate multiple testing correction is applied [80].

For common variants, association-based testing is used to test whether genotype predicts phenotype. For unrelated cases and controls, a Cochran–Armitage test for trend is often employed to test whether a specific genetic variant occurs more often in cases than controls. However, if covariates, such as age, sex, or adjustments for population stratification, need to be incorporated in the model, then logistic regression is used, with the variant re-coded as the number of minor alleles present. Use of genome-wide association (GWAS) has been reported for various CHD [81]. While most association studies utilize unrelated cases and controls, family-based association tests are based on the distribution of Mendelian transmissions from parents to their offspring [82]. While considered less powerful than using unrelated individuals, they are robust to population stratification [83] and are considered more powerful for rare variant analyses [84,85]. When testing for association with common variants, a thousand or more cases are often required with larger numbers of controls for reasonably well powered studies as effect sizes of common variants are often modest.

For rare variants, three approaches can be used: gene collapsing/burden tests, linkage analyses, and filtering. The gene collapsing method uses an association framework, but the variants are not considered individually, either by generating a binary score (presence or absence) or a continuous score. A common approach for gene collapsing is SNP-set (Sequence) Kernel Association Test (SKAT) which allows covariate adjustment [86]. While burden analyses provide evidence that variation in a gene is associated with CHD, it is not based on specific variants [87,88]. While simulations suggest that thousands of cases are required for well powered studies [89], work using burden analysis in CHD suggests that cases sample sizes below 1000 may be sufficient [87,88]. When trio data are avaialble, de novo varaints can also be identified. Notably, researchers can also evaluate whether specific genes or classes of de novo variants enriched in individuals with CHD compared to expectations based on probalistic modeling [90]. This approach has been used to demonstrate the importance of de novo variants in sydromic CHD, while non-syndromic CHD did not exhibit enrichment [91,92]. Similar to the gene collapsing methods, these approaches have been used with case sample sizes below 1000 [91,92]. Another approach for rare variant assessment is linkage analyses. Linkage analyses tests the hypothesis that the phenotype is inherited (segregates) with a variant more often than expected by chance. Early work in gene discovery for CHD used linkage analyses [68,93,94,95,96,97]. A challenge with linkage analyses is that the structure of the family impacts the power of discovery. Large families with multiple affected individuals across generations can be powerful as effect sizes are typically large and there is less background noise. But, such families may be difficult to find. Thus, many studies use many smaller families, but this can introduce genetic heterogeneity. It is important to recognize that linkage analyses leverage recombination events, and thus they identify a region rather than a variant. These regions may span millions of base pairs and harbor variants segregating with disease.

Beyond analytic approaches, many investigators simply use a process of prioritizing rare variants using a filtering approach first used by Ng and colleagues to evaluate exome sequencing data [98]. The filtering approach restricts variants based on the proposed inheritance model, population frequency, and the putative impact of a variant. With respect to inheritance, models include recessive (two copies of the alternative allele, or for compound heterozygotes two alternative alleles within a gene), dominant (one copy of the alternative allele), and de novo (a new mutation not found in the parents). For recessive and dominant inheritance, filtering can be done using cases only. Inclusion of unaffected parents (trios) can be used to further narrow the number of variants. Unaffected parents are required for detection of de novo rare variants. With respect to population frequency, the assumption is that a highly penetrant variant of a serious medical condition is not likely to be seen in the general population [99]. Aggregate databases like ExAC and gnomAD [100,101] provide access to large numbers of sequenced individuals. As these data were from studies primarily of adults, the assumption is that rare conditions with serious medical implications such as CHD are unlikely to be represented in the cohort. However, as noted above, conditions like BAV, which share genetic etiology with other CHD, often do not exhibit the morbidities until mid-late life, thus they could be in those datasets. Additionally, the use of ExAC and gnomAD is similar to using out of study controls, so care should be taken to ensure data comparability. Lastly, putative impact of variants is an important filtering criterion. Most NGS studies restrict variants to those that alter the protein and are bioinformatically predicted to impact protein function. Many tools have been developed to predict whether a protein change is likely to have functional impact [102,103,104,105]. It is important to recognize that these tools suffer from both false positives and negatives [106,107]. Further, the focus exclusively on variants which change protein structure ignores the potential role for regulatory variants. As heart development is a carefully orchestrated process where genes are turned on and off at the appropriate moments [108], it seems logical that regulatory variants could also contribute to CHD. The filtering strategy is an excellent approach for rare, highly penetrant conditions where the number of subjects is highly limited.

### 4.3. Implicating Variants in Disease Etiology

The ideal outcome of genetic studies is to identify variants that have a causal role, meaning they are necessary and sufficient for disease. However, as noted above, analytic approaches never achieve this goal. For CHD, which may be due to either multiple genetic hits [69] or genetic hits plus environmental insults [109,110,111], demonstrating causality may be an unattainable goal. A more realistic goal for genetic analytics is to identify variants in specific genes that influence occurrence of a phenotype which provide keys to open biologic doors that inform how the genetic variants modulate heart development. Further, we suggest that biologic studies should focus on evaluating the role of variants in CHD etiology rather than causality. The etiologic role can be explored with in vitro and/or in vivo studies. In vitro studies are powerful approaches for many conditions, but the evidence gained related to CHD should be interpreted with caution. This is because the developing heart is composed of both myocytes and endothelial cells and this cellular communication is critical for appropriate structural development [112]. In vivo models capture the cellular dynamics but there are still concerns for off-target effects with genome editing [113] and the potential differences in regulatory regions across species [114].

## 5. Clinical Considerations

Genetic testing of severe CHD has recently identified a genetic cause in only ~35% of the cases [115]. Having a genetic diagnosis can be useful clinically as it can help identify syndromic CHD which often requires different clinical management than non-syndromic CHD. Further, knowing the cause of CHD can help families understand their recurrence risk [116]. Given the familial data, it is likely that CHD will exhibit Mendelian inheritance in only a few families with highly penetrant variants and complex inheritance for most other families. These complex inheritance CHD will likely be missed by current clinical genetic testing as the focus is largely on highly penetrant rare variants. To address prediction in complex inheritance, use of polygenic risk scores has been proposed but the utility of such an approach is still uncertain [117,118]. However, we recognize that the long-term prognosis of CHD, even within specific types of CHD, is highly variable, thus the development of predictive models for outcomes, which may include biologic and genetic markers, rather than simply seeking to identify causes of CHD would be highly valuable. Ultimately, it is the hope that a genetic diagnosis can inform personalized medicine, but further work is required.

## 6. Conclusions

As recently reviewed [3], while considerable progress has been made in defining the genetic underpinnings of CHD, significant work remains. Available tools, such as high throughput genotyping and state-of-the-art analytic methods, will facilitate future studies advancing knowledge of CHD genetic etiology. However, it is becoming more evident that good genetic studies start with good planning. Continued progress in unraveling the genetic underpinnings of CHD will require multidisciplinary collaboration between geneticists, quantitative scientists, clinicians, and developmental biologists.

## Figures and Tables

**Figure 1 genes-12-00827-f001:**
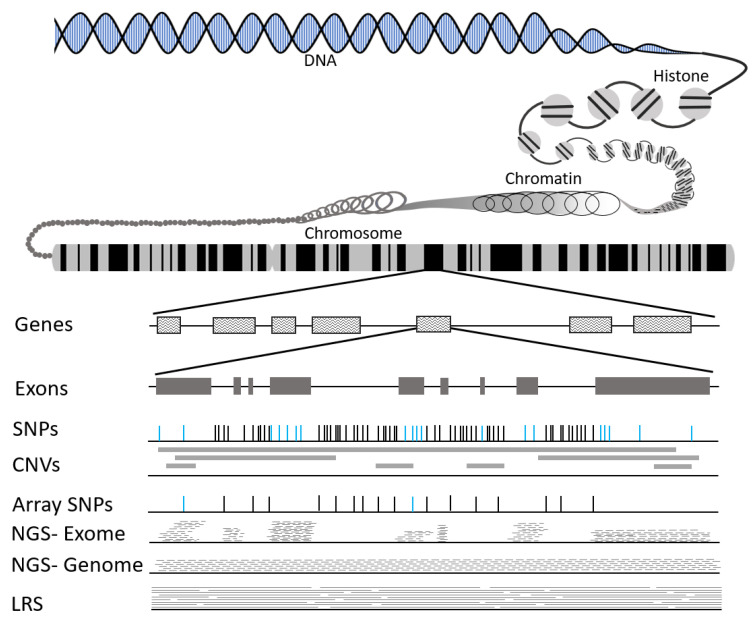
Illustration of DNA organization, types of variation present, and capture of variants using differing technologies.

**Table 1 genes-12-00827-t001:** Definition of genetic phenomena.

Phenomenon	Attribute
Genetic heterogeneity	Similar phenotypes, different genetic cause.
Variable expressivity	Individuals with same disease gene have different phenotypes.
Reduced penetrance	Disease absence in some individuals with disease gene.
Pleiotropy	Multiple phenotypes associated with the same genetic cause.

**Table 3 genes-12-00827-t003:** Study designs, analysis types, and limitations.

Study Design	Case Type	Control Type	Analysis Type	Limitations
Case Control	Unrelated	Unrelated -local	Association	Heterogeneity
		Unrelated—Out of study	Association	Heterogeneity Differences between cases and controls Differences in genotype generation
Trio	Unrelated	Parents of cases	Family Based Association (TDT)	Heterogeneity
			Linkage analyses	Heterogeneity
			Filtering	Generalizability
Family	Related—may be a series of families	Family members of cases	Family based Association	Heterogeneity
			Linkage analysis	Generalizability
			Filtering	Generalizability

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
