# Peer review of "Focused Strategies for Defining the Genetic Architecture of Congenital Heart Defects"

_genes, 2021, doi:10.3390/genes12060827_

Round 1
Reviewer 1 Report
Authors reported a narrative review with clear aim that was to elucidate strategies to improve our understanding of the genetic architecture of human CHD. The article is well balanced, logically structured and consists of important information for clinicans. Although the review covers several aspects of previous and current studies in the field of interests, I would like to recommend to extend section at the end of the paper adding critical comments for genes smart cards, genes predictive models, multiple biomarker models with genetic components, etc. It would be great if authors provide clear schema or figure illustrating an advance in their opinions regarding of whom, where and when in pre- and post-natal periods genes architecture yield benefits for the best.
Author Response
The Reviewer raises an excellent point. Our opinion is that the state of the field offers very little in the way of genetic testing of practical bedside use. While clinical testing can be useful in select “syndromic” cases, in general clinical genetic testing is limited because of the complex inheritance of CHD. We see many similarities between CHD and the problem of hypertension – rare cases are caused by single gene mutations, but genetic causes of more common types are poorly understood. As we recently reported (Teekakirikul et al.) identifying a role of common variants in rare disease may open doors to improved insight into genetic architecture.
To address this point, we have added a section entitled Clinical Considerations (line 360).
“Genetic testing of severe CHD has identified a genetic cause in only ~35% of cases [115]. Having a genetic diagnosis can be useful clinically as it can help identify syndromic CHD which often requires different clinical management than non-syndromic CHD. Further, knowing the cause of CHD can help families understand their recurrence risk [116]. Given the familial data, it is likely that CHD will exhibit Mendelian inheritance in only a few families with highly penetrant variants and complex inheritance for most other families. These complex inheritance CHD will likely be missed by current clinical genetic testing as the focus is largely on highly penetrant rare variants. To address prediction in complex inheritance, use of polygenic risk scores has been proposed but the utility of such an approach is still uncertain [117, 118]. However, we recognize that the long-term prognosis of CHD, even within specific type of CHD is highly variable, thus the development of predictive models for outcomes, which may include biologic and genetic markers, rather than simply seeking to identify causes of CHD would be highly valuable. Ultimately, it is the hope that a genetic diagnosis can inform personalized medicine, but further work is required.”
Reviewer 2 Report
This review article provides a comprehensive overview of CHD and its epidemiology and etiology, with a focus on current analytical and genomics methods that could be used to detect various types of genetic variations associated with congenital heart disease. The manuscript is well written so it’s a joy to read.
Minor comments
- Page 1: The first citation started from number 16. References should be reorganized.
- Line 79-80: It would be great if authors can discuss in more detail about syndromic and nonsyndromic cardiac phenotypes and how they can be classified.
- Line 160: The sentence is pretty confusing and needs to be rephrased. I assume the authors wanted to say, “WES provides less even and more biased coverage than WGS.”
- Table 2: Authors should add the legend to explain what “+” sign meant in “Laboratory Methods” and why “BAV/HLHS” was chosen to represent all CHD.
- Line 224-225: The names of those databases should be mentioned.
- Line 274: Filtering should be considered as a process of prioritizing mutations rather than an analytical method. The authors should discuss about the mutation rate-based expectation model used for de novo (PMID: 25086666, PMID: 26785492, PMID: 27479907), dominant and recessive rare variant analysis (PMID: 28991257, PMID:30409806) and their applications in CHD studies.
- Line 284: “exAC” should be “ExAC”.
- It would be great if authors can discuss how the cohort size and effect size of the variants impact the power of genetic analysis.
Author Response
This review article provides a comprehensive overview of CHD and its epidemiology and etiology, with a focus on current analytical and genomics methods that could be used to detect various types of genetic variations associated with congenital heart disease. The manuscript is well written so it’s a joy to read.
We appreciate the Reviewers encouraging remarks.
Minor comments
- Page 1: The first citation started from number 16. References should be reorganized.
Thank you. Corrected.
- Line 79-80: It would be great if authors can discuss in more detail about syndromic and nonsyndromic cardiac phenotypes and how they can be classified.
The Reviewer raises an excellent point which by itself could be the topic of a review. In the current review, we wish to emphasize that for genetic studies CHD phenotyping should extend beyond the heart. For example, from the perspective of an echocardiographer, tetralogy of Fallot due to del22q11 may be indistinguishable from that due to mutation in NKX2.5. However, in this instance genetic origins would be quite different. We have used this example and changed the text to emphasize this point. Line 77 now reads “For example, from the perspective of an echocardiographer, tetralogy of Fallot due to del22q11, aka DiGeorge syndrome, may be indistinguishable from that CHD due to NKX2.5 mutation, whereas in the same scenario, a history of cleft palate would drastically alter the genetic focus. [3]”
- Line 160: The sentence is pretty confusing and needs to be rephrased. I assume the authors wanted to say, “WES provides less even and more biased coverage than WGS.”
Our apologies. We have corrected the sentence.
- Table 2: Authors should add the legend to explain what “+” sign meant in “Laboratory Methods” and why “BAV/HLHS” was chosen to represent all CHD.
Thank you. Our apologies for the confusion. We have added a legend – “+ = effectiveness of method to capture.” We have also changed the heading to read “Examples in CHD.” In examples that follow, we mention examples of BAV/HLHS when appropriate and other times use the generic term “CHD”.
- Line 224-225: The names of those databases should be mentioned.
Corrected.
- Line 274: Filtering should be considered as a process of prioritizing mutations rather than an analytical method. The authors should discuss about the mutation rate-based expectation model used for de novo (PMID: 25086666, PMID: 26785492, PMID: 27479907), dominant and recessive rare variant analysis (PMID: 28991257, PMID:30409806) and their applications in CHD studies.
We have now noted that the filtering approach is a method to prioritize variants. “Beyond analytic approaches, many investigators simply use a process of prioritizing rare variants using a filtering approach first used by Ng and colleagues to evaluate exome sequencing data [98]. ”
We have also added the following “When trio data are avaialble, de novo varaints can also be identified. Notably, researchers can also evaluate whether specific genes or classes of de novo variants enriched in individuals with CHD compared to expectations based on probalistic modeling [90]. This approach has been used to demonstrate the important of de novo variants in sydromic CHD while non-syndromic CHD did not exhibit enrichment [91, 92].”
- Line 284: “exAC” should be “ExAC”. –
Corrected.
- It would be great if authors can discuss how the cohort size and effect size of the variants impact the power of genetic analysis.
At the end of the discussion of each analytic approach, we have added comments on the size of the cohort required for appropriately powered cohorts and expected effect sizes.
“When testing for association with common variants, a thousand or more cases are often required with larger numbers of controls for reasonably well powered studies as effect sizes of common variants are often modest.
While simulations suggest that thousands of cases are required for well powered studies [89], work using burden analysis in CHD suggests that cases sample sizes below 1000 may be sufficient [87, 88].
Similar to the gene collapsing methods, these approaches have been used with case sample sizes below 1000 [91, 92].
The filtering strategy is an excellent approach for rare highly penetrant conditions where the number of subjects is highly limited.”